# TSCMamba: Mamba Meets Multi-View Learning for Time Series Classification

## Abstract

Multivariate time series classification (TSC) is critical for various applications in fields such as healthcare and finance. While various approaches for TSC have been explored, important properties of time series, such as shift equivariance and inversion invariance, are largely underexplored by existing works. To fill this gap, we propose a novel multi-view approach to capture patterns with properties like shift equivariance. Our method integrates diverse features, including spectral, temporal, local, and global features, to obtain rich, complementary contexts for TSC. We use continuous wavelet transform to capture time-frequency features that remain consistent even when the input is shifted in time. These features are fused with temporal convolutional or multilayer perceptron features to provide complex local and global contextual information. We utilize the Mamba state space model for efficient and scalable sequence modeling and capturing long-range dependencies in time series. Moreover, we introduce a new scanning scheme for Mamba, called tango scanning, to effectively model sequence relationships and leverage inversion invariance, thereby enhancing our model's generalization and robustness. Experiments on two sets of benchmark datasets (10 datasets each) demonstrate our approach's effectiveness, achieving average accuracy improvements of 4.01-6.45% and 8.77% respectively over leading TSC models such as TimesNet and TSLANet.

## 1 Introduction

Time series classification (TSC) is a fundamental task in diverse fields abundant with time series data such as healthcare, weather forecasting, and finance. With the advancement of sensing technologies, multivariate time series (MTS) data have been ubiquitous, and thus TSC over MTS has attracted increasing research attention. Various approaches for TSC have been explored, including statistical, signal processing, and machine or deep learning approaches in both time- and frequency domains. Despite intensive studies and rapid progress, important properties of many time series, such as shift equivariance and inversion invariance, still remain underexplored in TSC.

Equivariance refers to a property where transformations applied to the input lead to predictable and corresponding transformations in the output. In the case of shift equivariance for time series data, this means that when we apply a shift transformation to the input time series, the output of our model (such as detected features or patterns) undergoes a corresponding shift. Shift equivariance property is crucial for TSC as it allows pattern recognition regardless of exact temporal positions, offering benefits such as:

- Resilience to temporal misalignments between different instances of the same class, which is common in real-world data.

- Creation of more consistent features across samples when dealing with MTS of varying lengths, such as when training and test examples have different durations.

- Improved generalization while retaining information about temporal relationships, as the classifier can learn patterns that are consistent in their relative positions across different time shifts.

These benefits collectively enhance the robustness and flexibility of TSC models, making them more adaptable to the diverse and often noisy nature of real-world time series data.

Existing machine learning approaches for TSC often rely on convolution-based methods or neural networks (CNNs) to endow shift equivariance for the extracted features (LeCun et al., 1989). Many models have been designed to exploit this property for visual learning tasks with 2D or 3D data (Thomas et al., 2018; Fuchs et al., 2020). For TSC, several convolution-based methods (Dempster et al., 2020; Franceschi et al., 2019; Ismail Fawaz et al., 2019) have implicitly utilized this property. However, they only employ temporal features by applying convolutions in the time-domain.

In addition to time-based features, frequency-based features can help identify important patterns in the data's spectral composition. Disentangling time and frequency components has been shown to be critical for time series learning (Zhang et al., 2022b; Eldele et al., 2024). In particular, spectral features have been extracted using discrete Fourier transform (DFT) (Eldele et al., 2024) and discrete wavelet transform (DWT) (Chaovalit et al., 2011; Ouyang et al., 2021; Zhao & Zhang, 2005). However, neither DFT nor DWT is shift equivariant - DFT lacks this property (Oppenheim, 1999), while DWT's discrete nature and downsampling (decimation) step in its computation also prevent shift equivariance (Mallat, 2008; Percival & Walden, 2000). Consequently, a small shift in the input signal can lead to significantly different coefficients or features for DFT and DWT.

Continuous wavelet transforms (CWT) with real-valued mother wavelets possess shift equivariance (Mallat, 2008). CWT coefficients offer a localized time-frequency representation of the signal, with scale-dependent locality (Mallat, 2008). This means they capture more localized features at higher frequencies and broader features at lower frequencies, while still maintaining overall locality compared to global transforms like DFT. While CWT has been used for TSC (Wang et al., 2020), global patterns of MTS data have been insufficiently considered. MTS data often contain sensible global patterns, such as trends, seasonality, periodicity, cycles, and long-term dependencies. Lacking the ability to capture shift-equivariant global features can lead to a loss of discriminative information for TSC.

These considerations highlight that many existing approaches cannot fully exploit shift equivariance of features or patterns for TSC. While some CNN or CWT-based methods can capture shift-equivariant features, they often lack the ability to effectively utilize spectral or global features. A clear need arises for an effective approach that can leverage shift-equivariant local and global features in both time and frequency domains to enhance TSC performance. In this paper, we propose a novel approach for TSC that effectively leverages time shift-equivariant features and patterns from both time and frequency domains. The shift-equivariant spectral features are derived from the time-frequency representations of CWT with a real-valued mother wavelet. To remedy CWT's limitation regarding the locality of resulting features, we leverage convolutional kernels to extract shift-equivariant temporal features.

While convolutional kernels excel at capturing temporal dependencies between input features or interdependencies between channels preserving shift equivariance, they typically have limited lengths, resulting in a limited receptive field that extracts temporally localized contents (Bengio et al., 2013). To enrich the expressiveness of CNN-based features, we adopt the kernel-based feature transformation ROCKET (Dempster et al., 2020). ROCKET uses random kernels with random lengths, potentially extracting a wide spectrum of temporal features, ranging from highly local to global ones.

However, many TSC tasks involve MTS data with characteristics or patterns spanning the entire length or a significant portion of the time series, suggesting that globally discriminative features or global interactions of features can be beneficial. While ROCKET may capture global features to a certain extent due to its use of kernels with random lengths, it tends to focus more on local features with varying degrees of locality. To enhance the extraction of global features, we incorporate fully connected MLPs. These can capture temporally global patterns with their wide receptive field, although they may not be well-suited for identifying temporally local patterns or temporal dependencies due to their treatment of each input feature independently. Thus, we use MLPs to complement ROCKET's approach, strengthening the extraction of global features.

To avoid significantly increasing the size of extracted features, we propose a switch mechanism that selects between CNN-based (primarily local) features and MLP-based global patterns. This mechanism determines which type of temporal features - local or global - is more discriminative for a given input and integrates it with the CWT-domain features. Our experiments demonstrate that the switch mechanism effectively captures the most salient characteristics of the input data, whether they are local or global in nature.

The time-frequency representations from CWT and temporal features from ROCKET transformation, or temporal features from global MLP, will be leveraged jointly to exploit the MTS characteristics, particularly shift equivariance, across domains. These different types of features provide various perspectives on the MTS data. We combine these perspectives with an approach known as multi-view learning, which has been shown to improve model performance and reliability, e.g., in image classification and clustering (Peng et al., 2024). This provides comprehensive, enriched multi-view contexts to the subsequent inference, thus enhancing TSC performance.

In addition, we introduce inversion invariance, a new concept in TSC where a time series' features or patterns are equally useful for classification when read in both forward and backward directions. This property is particularly relevant for data where time direction is not inherently meaningful, such as ECG patterns, climate data, or rotational data with arbitrary start points. We posit that inversion invariance can generally enhance TSC performance for the following reasons:

- Using inversion invariance for TSC can effectively double the amount of input data, as both forward and backward readings will be used to train the same model. This increase in training examples is a new form of data augmentation, which leads to better generalization, reducing potential overfitting for TSC.

- Capturing inversion invariant patterns may help enhance the model's robustness to noise or disturbances. MTS data is often noisy (Kang et al., 2014), which can mask underlying signals and affect algorithm robustness. By identifying patterns meaningful in both directions, the TSC model may focus more on intrinsic patterns while being less affected by (potentially direction-specific) noise.

Inversion invariance property may improve generalization and robustness, especially in cases where patterns can manifest similarly important patterns in both time directions. To incorporate this property into our model, we will leverage a new scanning scheme called tango scanning, which we will introduce below. Our empirical results based on extensive experiments on various MTS datasets testify to the usefulness of this property and thus support the postulate, though a theoretical certificate for upholding the postulate remains a future line of research.

To facilitate final classification, we employ Mamba (Gu & Dao, 2023), a state-of-the-art (SOTA) model based on state-space models (SSMs). Like recurrent neural networks, SSMs use state variables to represent the system's internal condition and its evolution over time (Gu et al., 2021). Mamba introduces selective state spaces, a mechanism that updates only a subset of state dimensions based on each input. This allows Mamba to focus on the most relevant information, efficiently process long sequences, and capture long-range dependencies. Mamba-based models have demonstrated competitive performance on various tasks, including language modeling (Gu & Dao, 2023; Dao & Gu, 2024), time series forecasting (Ahamed & Cheng, 2024b), DNA sequence modeling (Gu & Dao, 2023), tabular data learning (Ahamed & Cheng, 2024a), and audio generation (Shams et al., 2024). Unlike popular Transformers, which have quadratic time complexity in sequence length, Mamba achieves linear time complexity. This makes it more suitable for processing long sequences and scaling to larger datasets. By using Mamba, our TSC model is efficient in training and inference, with reduced computational costs and memory requirements compared to existing SOTA models.

We introduce a novel sequence scanning scheme, tango scanning, for inputting the original sequence and the reversed sequence into the same Mamba block and fusing the output. This scheme uses essentially the same memory footprint but demonstrates higher accuracy than vanilla Mamba scanning. Our tango scanning scheme differs from existing bi-directional Mamba implementations in the following ways: 1) Compared to (Wang et al., 2024; Behrouz & Hashemi, 2024), our approach uses one vanilla Mamba block for both directions, while theirs use two separate blocks. 2) Compared to BiMamba (Schiff et al., 2024), we use a

single Mamba block without weight ties, and only one reversal operation, whereas BiMamba uses two blocks with partial weight ties and two reversal operations. 3) Compared to MambaDNA (Schiff et al., 2024), we use a single reversal operation and one Mamba block, while MambaDNA uses two "reverse complement" operations and two Mamba blocks with weight ties where the "reverse complement" operation is specific to deoxyribonucleic acid (DNA) sequences to reflect the complementary nucleotides in the DNA base pairs. Through extensive experiments, we demonstrate that our multi-view Mamba-based approach outperforms or matches the performance of existing SOTA models, typically with a small fraction of computational requirements and reduced memory usage.

In summary, the contributions of this paper include:

- Building a novel multi-view approach for TSC: Our approach seamlessly integrates frequency- and time-domain features to exploit shift-equivariant patterns and provide complementary, discriminative contexts for TSC. It also employs a gating scheme to fuse spectral features with local or global time-domain features, effectively leveraging patterns characterizing the MTS classes.

- Adapting the Mamba state-space model for sequence modeling and TSC: Our model can capture long-term dependencies within the MTS with linear efficiency and scalability.

- Introduction of inversion invariance for TSC: This includes the new concept of inversion invariance for MTS classification. It also includes building an innovative Mamba-based "tango scanning" scheme to identify inversion invariant features or patterns. Our proposed tango scanning demonstrates improved effectiveness in modeling inter-token relationships in the sequence compared to vanilla Mamba block or bi-directional Mamba for TSC.

- Extensive experimental validation of the proposed approach: Our model shows superior performance over various existing SOTA models over two sets of standard benchmarking datasets (10 datasets each), achieving average accuracy improvements of 4.01-6.45% and 8.77% respectively over leading TSC models among 20 SOTA baselines.

These contributions are expected to advance real-world TSC applications in research and everyday life. The following sections will briefly review related works, present our approach in detail, and demonstrate extensive experimental results and ablation studies to conclude the paper.

## 2 Related Works

In this section, we provide a brief review of relevant methods for TSC in the literature, focusing on works using machine learning or deep learning. We group existing methods into 4 categories: traditional methods like DTW, deep learning approaches using CNNs or RNNs, Transformer architectures, and methods based on state-space models.

Traditional TSC methods include techniques like Dynamic Time Warping (DTW) (Berndt & Clifford, 1994), which measures the similarity between time series by aligning them in a non-linear way of dynamic programming. Tree-based methods like XGBoost (Chen & Guestrin, 2016) have also been applied to the TSC task.

In recent years, deep learning approaches have become increasingly popular for TSC. Various MLP-based methods have been proposed, including DLinear by Zeng et al. (2023) and LightTS by Zhang et al. (2022a). DLinear constructs a simple model based on MLP, while LightTS uses light sampling-oriented MLP. These models are generally efficient in computations. Convolutional neural networks (CNNs) have been adapted for TSC, such as ROCKET (Dempster et al., 2020) which uses random convolutional kernels for fast and accurate classification. CNN has been also used by Franceschi et al. (2019) for learning representations of multivariate time series in an unsupervised way, which is then further leveraged for TSC. Besides CNNs, recursive neural networks (RNNs) such as long-short-term memory (LSTM) (Hochreiter & Schmidhuber, 1997) and a variant, a gated recursive unit (GRU), has also been adopted for TSC. Moreover, CNN and RNN have been combined to handle TSC effectively (Lai et al., 2018).

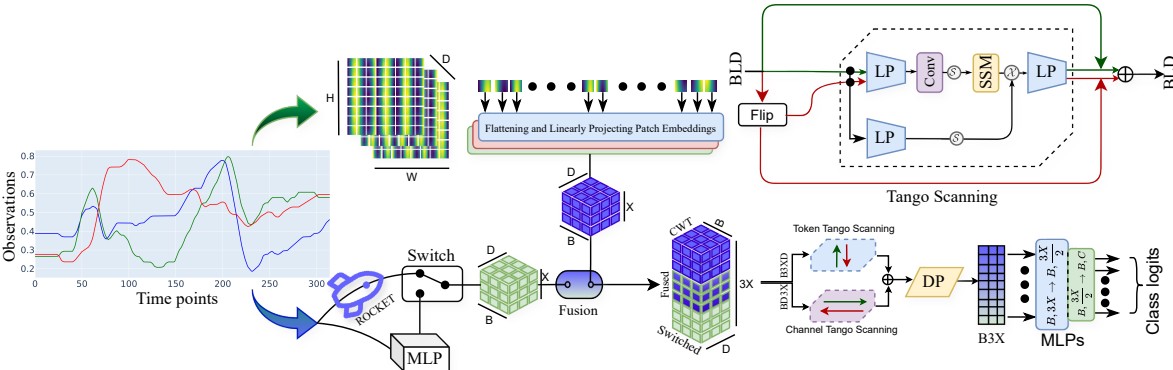

Figure 1: Schematic illustration of the proposed model TSCMamba. This diagram illustrates the architecture of our approach, featuring tango scanning. Here, DP refers to Depth-wise Pooling (DP) and LP refers to Linear Projection (LP). A switch gate selectively activates the utilization of either ROCKET or MLP-derived features. The MLP module, depicted in the bottom right, comprises two layers with an optional dropout mechanism interspersed for regularization

Transformers by Vaswani et al. (2017), originally used for natural language processing, have been adapted for time series modeling. Reformer by Kitaev et al. (2020) introduces efficiency improvements to handle longer sequences. Numerous Transformer variants have been proposed to better model the unique characteristics of time series, such as handling non-stationarity with on-stationary Transformers by Liu et al. (2022), combining exponential smoothing with ETSformer in Woo et al. (2022), and using decomposition and auto-correlation with Autoformer in Wu et al. (2021). Other variants include Pyraformer (Liu et al., 2021), which uses pyramidal attention to reduce complexity, Flowformer (Wu et al., 2022), which linearizes Transformers using conservation flows, and Informer (Zhou et al., 2022), which focuses on efficient long sequence forecasting by exploiting frequency-enhanced decomposition. Notably, TimesNet (Wu et al., 2023) models the temporal 2D variations in time series data using a hierarchical structure of temporal blocks. By combining 2D convolutions, multi-head self-attention, and a novel positional encoding scheme, it can capture both local patterns and long-range dependencies in time series and obtain state-of-the-art performance. Despite the impressive performance of Transformer-based models, Zeng et al. (2023) have shown that MLP-based models can be more effective in many scenarios.

Recently, a method called LSSL by Gu et al. (2022) has been proposed for TSC with a structured SSM called S4. It employs a special parameterization called the Diagonal Plus Low-Rank form to represent the state transition matrix, enabling efficient computation over long sequences.

## 3 Methodology

This section describes our proposed method, TSCMamba, whose overall architecture is schematically illustrated in Fig. 1. The architecture and data flow comprise the following key steps: First, we generate global temporal features, localized temporal features, and joint temporal-spectral features from the MTS data. Second, these diverse views are fused to provide rich contexts for subsequent sequence modeling. A switch gate determines whether to fuse the transform-domain features with local features or global patterns in the temporal domain. Third, both the pre-fusion and post-fusion features are fed into an inference engine consisting of two Mamba blocks. This captures long-term dependencies between these features. Fourth, we introduce a tango scanning scheme for each Mamba block to exploit inversion-invariant features, followed by depth-wise pooling. Fifth, final class decisions are made using an MLP to generate class logits. The following subsections detail each component and procedure of TSCMamba.

### 3.1   Spectral Representation

We have chosen Continuous Wavelet Transform (CWT) to represent raw signals in the spectral domain. CWT potentially surpasses space-time Fourier transform, fast Fourier transform (FFT), and digital wavelet transform (DWT) by providing superior time-frequency localization and multi-resolution analysis. CWT's adaptable wavelets enhance feature extraction and noise reduction while better-handling edge effects. This makes CWT particularly suitable for analyzing non-stationary signals. CWT's continuous and detailed representation may offer a significant advantage over the discrete nature of DWT. This renders CWT highly effective for precise time-frequency analysis.

Among a variety of wavelets, the Morlet wavelet in Equation 1 is employed in this paper due to its capturing both amplitude and phase effectively:

$$\psi(t) = (\pi^{-1/4})(1 - \frac{t^2}{\sigma^2}) \exp(-\frac{t^2}{2\sigma^2}) \cos(2\pi f t), \tag{1}$$

where $\sigma$ is the scale parameter controlling the width of the wavelet, and $f$ is the frequency parameter that controls the frequency of the cosine function. In this paper, we adopt $\sigma^2 = 1$ and $f = 5/(2\pi)$ to balance computational cost and the expressiveness of the obtained wavelet features. However, keeping these parameters learnable may potentially benefit the classification accuracy. The smooth and symmetric shape of the Morlet wavelet minimizes distortions and edge effects, resulting in a clear and interpretable time-frequency representation. Using the wavelet function, we obtain a 2-D representation of the size $L_1 \times L_1$ for each channel of an original MTS input sample of size $L$. In this paper, we adopt $L_1 = 64$ for computational efficiency and expressiveness of the obtained wavelet features. We summarize this CWT feature extraction process in S-Algorithm 1. Since conversion from time signals to CWT representation is not learnable, we move this to the data pre-processing part, while regarding only the patch embedding module to be learnable. This helps our model to achieve lower FLOPs and faster training.

With the resultant CWT representation of size $D \times L_1 \times L_1$, we further perform patch embedding using a Conv2D layer (kernel size = stride = $p$, padding = 0), where $p = 8$ is patch size. Later with flattened patches, we utilize a feed-forward network (FFN) to obtain patches of size $D \times X$ for each MTS sample. The FFN consists of one fully connected layer with an input dimension $(\frac{L_1}{p})^2$ and an output dimension $X$, as shown for the projected space in Figure 1. It is used to extract features within the CWT representation. For each batch of size $B$, the resultant tensor for representing CWT features is denoted by $\mathcal{W} \in \mathcal{R}^{B \times D \times X}$.

### 3.2   Temporal Feature Extraction

To complement the frequency-domain features, we extract time-domain features. As previously discussed, different MTS datasets may have global or local features or patterns that discriminate between different classes. Capturing these features is essential for accurate classification. We leverage two different approaches to capture such features.

**Extracting Local Features with Convolutional Kernels in Unsupervised Fashion.** Convolutional kernels usually have limited receptive fields, thus focusing on the extraction of local features. Since an MTS dataset may have local features at multiple temporal scales, it is sensible to capture local features within various widths of receptive fields. To this end, we employ the ROCKET approach (Dempster et al., 2020) to extract local features within various local neighborhoods for each channel in an unsupervised fashion. Here, it is to be noted that we only utilize the time domain to extract the kernel-based features, we do not utilize the class labels of the corresponding features. Therefore, our improvement in performance does not solely rely on the ROCKET method, rather it works as a performance booster in certain datasets.

ROCKET is a randomized algorithm that uses a set of randomized convolutional kernels to extract features from time series data. The method is suitable for capturing local features at various scales due to its randomized nature and the use of kernels with different sizes and strides. The procedure first randomly generates a set of convolutional kernels, each with a specific size and stride. Next, it convolves each kernel with the time series data to generate a feature map. The procedure is summarized in S-Algorithm 2. ROCKET generates random convolutional kernels, including random length and dilation. It transforms the

time series with two features per kernel. The global max pooling and the proportion of positive values (PPV). This fits one set of parameters for individual series.

We apply the ROCKET feature extraction method to each channel of length $L$ to form a feature vector of length $X$. We input our training data as a result, with the input tensor of size $B \times D \times L$, we obtain a tensor $\mathcal{V}_L \in \mathcal{R}^{B \times D \times X}$ that represents the local features. We utilized the sktime implementation of ROCKET to achieve this (Király et al., 2024; Löning et al., 2019).

**Global Feature Extraction with MLP.** MLP has a receptive field covering the entire input, allowing the resulting feature vectors to capture the global characteristics of the MTS data. Independently for each channel of the input MTS of size $D \times L$, we utilize a one-layer MLP with linear activation to obtain a feature vector of size $D \times X$. Therefore, with the input tensor of size $B \times D \times L$, we obtain a tensor $\mathcal{V}_G \in \mathcal{R}^{B \times D \times X}$ representing the global features.

### 3.3 Fusing Multi-View Representations

After obtaining the CWT-based spectral features using CWT and the temporal features at both local and global levels, we fuse these features to effectively exploit the complementary information in these multi-view representations. Through our empirical study, we observe that for many MTS data, either local features or global features in the temporal domain play a dominant role in discriminating between classes for TSC. This observation motivates us to fuse the spectral features with either global or local features in the temporal domain. Denote the temporal features by $\mathcal{V}$, which is either $\mathcal{V}_G$ or $\mathcal{V}_L$. Then, the fused feature map $\mathcal{V}_W$ will be calculated as follows:

$$\mathcal{V}_W = \mathcal{W} \otimes \mathcal{V}, \tag{2}$$

where $\otimes$ represents an element-wise operation. In this paper, this is either a multiplicative or additive operation such that

$$\{\mathcal{V}_W\}_{ijk} = \lambda \mathcal{V}_{ijk} * (2 - \lambda)\mathcal{W}_{ijk}, \quad \text{or,} \quad \{\mathcal{V}_W\}_{ijk} = \lambda \mathcal{V}_{ijk} + (2 - \lambda)\mathcal{W}_{ijk}, \tag{3}$$

where $\lambda \geq 0$ is a learnable parameter that determines the balance between the spectral features and the temporal features, $1 \leq i \leq B$, $1 \leq j \leq D$, $1 \leq k \leq X$. We set the $\lambda$ as a learnable parameter while the initial value of $\lambda = 1.0$. Therefore, the optimal value of $\lambda$ will be determined during the training process. The initial value of $\lambda = 1.0$ ensures a balanced focus initially between temporal and spectral domain features.

After obtaining the fused temporal-spectral features, we composite it with the tensors containing the multi-view features into a new tensor $\mathcal{U} = \mathcal{W} \| \mathcal{V}_W \| \mathcal{V} \in \mathcal{R}^{B \times D \times 3X}$, where $\|$ is a concatenation operation. We use a switching mechanism to make the choice between $\mathcal{V} = \mathcal{V}_G$ or $\mathcal{V} = \mathcal{V}_L$. This mechanism is implemented as a learnable binary mask that selects either the global or local temporal features during the training process. The final state of this switch will be determined by the optimization in the training process and tuned based on datasets for optimal performance.

### 3.4 Inferring with Time-Channel Tango Scanning

With the integrated temporal-spectral contextual representations contained in tensor $\mathcal{U}$, which is processed by a layer normalization for training stability, we can now learn salient representations to capture important relationships between features, particularly long-term dependencies. To achieve this, we construct tokens by treating each feature vector in $\mathcal{U}$ along the time and channel dimensions as a separate token. Subsequently, we leverage Mamba, a type of SSM, for modeling the token sequences. Mamba is designed for capturing discriminative contents by selectively scanning the token sequences. This selective scan ability allows the model to focus on the most informative parts of the sequence while ignoring less relevant information. By doing so, Mamba can effectively capture long-term dependencies and identify salient features that are most useful for classification.

Compared to other SSMs, Mamba has the advantage of being computationally efficient and able to handle long sequences. It achieves this by using a sparse attention mechanism that reduces the complexity of token-to-token interactions. This makes Mamba particularly well-suited for processing time series data, where the sequences can be lengthy and contain complex temporal dependencies.

**Vanilla Mamba Block:** Inside a Mamba block, two fully-connected layers in two branches calculate linear projections. The output of the linear mapping in the first branch passes through a 1D causal convolution and SiLU activation $\mathcal{S}(\cdot)$ (Elfwing et al., 2018), then a structured SSM. The continuous-time SSM maps an input function or sequence $u(t)$ to output ~~$v(t)$~~ $z(t)$ through a latent state $h(t)$:

$$dh(t)/dt = A \, h(t) + B \, u(t), \quad z(t) = C \, h(t), \tag{4}$$

where $h(t)$ is $N$-dimensional, with $N$ also known as a *state expansion factor*, $u(t)$ is $D$-dimensional, with $D$ being the *dimension factor* for an input token, $z(t)$ is an output of dimension $D$, and $A$, $B$, and $C$ are coefficient matrices of proper sizes. This dynamic system induces a discrete SSM governing state evolution and outputs given the input token sequence through time sampling at $\{k\Delta\}$ with a $\Delta$ time interval. This discrete SSM is

$$h_k = \bar{A} \, h_{k-1} + \bar{B} \, u_k, \quad z_k = C \, h_k, \tag{5}$$

where $h_k$, $u_k$, and $z_k$ are respectively samples of $h(t)$, $u(t)$, and $z(t)$ at time $k\Delta$,

$$\bar{A} = \exp(\Delta A), \quad \bar{B} = (\Delta A)^{-1}(\exp(\Delta A) - I)\Delta B. \tag{6}$$

For SSMs, diagonal $A$ is often used. Mamba makes $B$, $C$, and $\Delta$ linear time-varying functions dependent on the input. In particular, for a token $u$, $B, C \leftarrow Linear_N(u)$, and $\Delta \leftarrow softplus(parameter + Linear_D(Linear_1(u)))$, where $Linear_p(u)$ is a linear projection to a $p$-dimensional space, and $softplus$ activation function. Furthermore, Mamba also has an option to expand the model dimension factor $D$ by a controllable dimension expansion factor $E$. Such coefficient matrices enable context and input selectivity properties (Gu & Dao, 2023) to selectively propagate or forget information along the input token sequence based on the current token. Denote the discretization operation by $\Delta = \tau_\Delta(parameter + s_\Delta)$, where $\tau_\Delta$ and $s_\Delta$ are both functions of the input. For the special case of univariate sequences, the selectivity property has been mathematically proved (Gu & Dao, 2023), as shown in the following:

**Theorem 1.** *(Gu and Dao 2023) When $N = 1, A = -1, B = 1$, $s_\Delta = Linear(x)$, and $\tau_\Delta = softplus$, then the selective SSM recurrence takes the form of*

$$h_k = (1 - g_k) \, h_{k-1} + g_k \, u_k, \quad and \quad g_k = \sigma(Linear(u_k)), \tag{7}$$

*where $g_k$ is the gate.*

This theorem states that the hidden state is a convex combination of the current input token and the previous hidden state, with the combination coefficient controlled by the current input token. Moreover, it is pointed out that the parameter $g_k$ is responsible for selecting the input contents $u_k$ from the sequence, plays a role similar to a gating mechanism in the RNN model, thus connecting the selective SSM to the traditional RNN.

After obtaining the SSM output, it is multiplicatively modulated with the output from the second branch before another fully connected projection. The second branch in the Mamba block simply consists of a linear mapping followed by a SiLU.

**Tango Scanning:** The selectivity ability of Mamba depends on the ordering of the tokens in the sequence because the hidden state at time $n$ is constructed causally from history tokens as determined by the ordering of the tokens. If the history tokens do not contain informational contexts, Mamba may provide less effective predicted output. To alleviate this potential limitation of causal scanning, we construct a dedicated module to extend a vanilla Mamba block, as shown in Figure 1. Each module comprises one vanilla Mamba block. On the input side, the module accepts a sequence in a forward fashion as input and then inverts the sequence to accept it as input again. At the output side, the output of the vanilla Mamba block with forward sequence and that with the inverted sequence are added element-wise. The operations are represented as follows. Denote an input token sequence by $v = [v_1, \cdots, v_M]$, where $v_i \in \mathcal{R}^D$, and $v \in \mathcal{R}^{D \times M}$ is the matrix representation of the token sequence with $M$ being the sequence length. We will first get a reverse-flipped sequence $v^{(r)}$ by inverting the ordering of the elements in $v$. Tango scanning performs the following operations to obtain the output sequence $s^{(o)}$:

$$v^{(r)} = Reverse(v) = [v_M, v_{M-1}, \cdots, v_1], \tag{8}$$

$$a = Mamba(v), \quad a^{(r)} = Mamba(v^{(r)}), \tag{9}$$

$$s^{(o)} = v \oplus a \oplus v^{(r)} \oplus a^{(r)}, \tag{10}$$

where $Reverse(\cdot)$ denotes the flipping operation of a sequence, $Mamba(\cdot)$ denotes a vanilla Mamba block, and $\oplus$ denotes element-wise addition. The last equation 10 represents the element-wise addition for information fusion. Notably, the same Mamba block is used for the forward sequence $v$ and the reverse-flipped sequence $v^{(r)}$. By doing so, the SSM in this block will be trained to update the hidden state variable more effectively than using simply the forward scanning of the vanilla Mamba. Because of the sharing of one Mamba block (and thus one SSM) with two sequences that are flips of each other, we regard it as a dancer's one body with two concerted legs and hence call it tango scanning.

Unlike the bi-directional Mamba block in Behrouz & Hashemi (2024); Schiff et al. (2024) that uses two separate SSMs with one for forward direction and the other for backward direction, our tango scanning block uses only a single Mamba block. Importantly, for the inverted sequence in our tango scanning module, the output from the Mamba block is not re-inverted back to the original order before performing element-wise addition. In other words, a tango scanning module only involves sequence inversion once. On the contrary, the bi-directional Mamba block (Schiff et al., 2024) needs to re-invert the output from the vanilla Mamba block. Empirically, we will demonstrate that our tango scanning can effectively update the hidden state variable while maintaining essentially the same memory footprint as the vanilla Mamba block.

**Performing Tango Scanning in Time and Channel Dimensions:** The MTS data have significant patterns, correlation structures, and temporal long-term dependencies. To model the relationship in the temporal dimension, we perform tango scanning temporally for every channel. The processed embedded representation with tensor size $B \times 3X \times D$ is transformed using Tango Scanning. Specifically, with each $D$-dimensional feature point across all channels regarded as a token, we have a token sequence with dimension factor $D$ and length $3X$ as input to the Mamba block in the tango scanning module. This yields an output tensor of size $B \times 3X \times D$. That is, by denoting $u^{(t)} = [u_1^{(t)}, \cdots, u_{3X}^{(t)}]^T \in \mathcal{R}^{3X \times D}$ as the token sequence formed along the time direction for a time series (in the batch), we have

$$s^{(t)} = Tango\_Scanning(u^{(t)}), \tag{11}$$

where $s^{(t)} = [s_1^{(t)}, \cdots, s_{3X}^{(t)}]^T \in \mathcal{R}^{3X \times D}$. By leveraging Mamba, we will extract salient features and context cues from the input token sequence. Particularly, the output sequence $s_t$ captures the between-time-point interactions along the temporal direction.

Because the MTS data often have significant correlations along the channel dimension, we will also model relationships across channels. To this end, we first form our tensor to have size $B \times D \times 3X$ and then we transform it using our tango scanning. Specifically, the whole univariate sequence of each channel is used as a token with a dimension factor $3X$ for the Mamba block in the tango scanning module. Thus, we form a token sequence of length $D$, with each token having dimension $3X$. This token sequence will be input to our tango scanning module, yielding an output tensor of size $B \times D \times 3X$. That is, by denoting $u^{(c)} = [u_1^{(c)}, \cdots, u_D^{(c)}]^T \in \mathcal{R}^{D \times 3X}$ as the token sequence formed along the channel dimension, we have

$$s^{(c)} = Tango\_Scanning(u^{(c)}), \tag{12}$$

where $s^{(c)} = [s_1^{(c)}, \cdots, s_D^{(c)}]^T \in \mathcal{R}^{D \times 3X}$. Note that the Tango Scanning module used in Eq. (12) is different from the one used in Eq. (11) and utilizes a separate Mamba module. The output sequence $s_c$ captures the between-channel interactions along the temporal direction. It is critical to account for the inter-relationships across channels when the MTS data have many channels.

After obtaining the outputs from the time-wise scanning $u_t$ and the channel-wise scanning $u_c$, we will perform another fusion at the Mamba-transformed sequence level:

$$z = (s^{(t)})^T \oplus s^{(c)}, \tag{13}$$

where $\oplus$ denotes element-wise addition of matrices, and $(s^{(t)})^T$ is the transpose of $s^{(t)}$. The resultant fused sequence is of size $D \times 3X$.

### 3.5 Output Class Representation

The fused tensor of size $B \times D \times 3x$ will be used to distill class information (class logits). First, we perform depth-wise pooling (DP) (Figure 1) to aggregate information across channels. Specifically, given a fused

sequence $z \in \mathcal{R}^{D \times 3X}$, we have

$$\bar{z} = DP(z), \tag{14}$$

where $DP(\cdot)$ denotes the depth-wise pooling and the output $\bar{z} \in \mathcal{R}^{3x}$. DP can be either average pooling or max pooling. We regard these two pooling operations as two possible values of DP. Given a dataset, the specific pooling will be determined in the training stage.

Subsequently, we will employ an FFN of two layers with an optional dropout mechanism interspersed for regularization:

$$\bar{z}^{(1)} = MLP(\bar{z}), \quad \bar{z}^{(2)} = MLP(\bar{z}^{(1)}), \tag{15}$$

where $\bar{z}$ is projected into vectors $\bar{z}^{(1)} \in \mathcal{R}^{3x/2}$ and $\bar{z}^{(2)} \in \mathcal{R}^{C}$. The class labels will be determined based on $\bar{z}^{(2)}$. To train the proposed network, which we call TSCMamba, we employ a cross-entropy loss (CE) on the output of the second layer of the FFN, $\bar{z}^{(2)}$.

## 4 Experiments and Result Analysis

In this section, we present the results of our experiments on benchmark datasets for time series classification tasks. We evaluate the performance of our proposed method, TSCMamba, and compare it with several state-of-the-art baseline models. The results demonstrate the effectiveness of TSCMamba in handling complex time series classification tasks.

### 4.1 Datasets

We evaluated the performance of our proposed method, TSCMamba, on 10 benchmark datasets for time series classification tasks following TimesNet (Wu et al., 2023) and additional 10 datasets following TSLANet (Eldele et al., 2024). These datasets are commonly used in the literature and are representative of various domains, including image, audio, and sensor data. We present dataset statistics in S-Table 1. These datasets are sourced from a diverse set of domains and contain a diverse range of classes, channels, and time-sequences leading to a robust benchmark for evaluating classification tasks. Moreover, some datasets contain more data in the Test set than the Train set (EC, HW, HB, JV, SCP1, UG) making the time-series classification a harder task. More domain-related information can be found in Bagnall et al. (2018).

### 4.2 Experimental Environment

All experiments were conducted using the PyTorch framework (Paszke et al., 2019) with NVIDIA $4\times$ V100 GPUs (32GB each). The model was optimized using the ADAM algorithm (Kingma & Ba, 2014) with Cross-Entropy loss following TimesNet (Wu et al., 2023). Moreover, the baseline results are utilized from TimesNet (Wu et al., 2023) paper for a fair comparison (Same train-test set across the methods). The batch

Table 1: Classification Accuracy (%) for Various Models. The . symbol in Transformer models denotes the specific type of $*$former used. The best average result and rank is in **bold** and second best is underlined. The ranks are calculated using the Wilcoxon signed-rank test (lower is better).

| Datasets | DTW (1994) | XGBoost (2016) | Rocket (2020) | LSTM (1997) | LSTNet (2018) | LSSL (2022) | TCN (2019) | Trans (2017) | Re. (2020) | In. (2021) | Pyra (2021) | Auto. (2021) | Station. (2022) | FED. (2022) | ETS. (2022) | Flow. (2022) | DLinear (2023) | LightTS (2022a) | TimesNet (2023) | TSLANet (2024) | TSCMamba Ours |
|---|---|---|---|---|---|---|---|---|---|---|---|---|---|---|---|---|---|---|---|---|---|
| EC | 32.3 | 43.7 | 45.2 | 32.3 | 39.9 | 31.1 | 28.9 | 32.7 | 31.9 | 31.6 | 30.8 | 31.6 | 32.7 | 31.2 | 28.1 | 33.8 | 32.6 | 29.7 | 35.7 | 30.4 | 62.0 |
| FD | 52.9 | 63.3 | 64.7 | 57.7 | 65.7 | 66.7 | 52.8 | 67.3 | 68.6 | 67.0 | 65.7 | 68.4 | 68.0 | 66.0 | 66.3 | 67.6 | 68.0 | 67.5 | 68.6 | 66.8 | 69.4 |
| HW | 28.6 | 15.8 | 58.8 | 15.2 | 25.8 | 24.6 | 53.3 | 32.0 | 27.4 | 32.8 | 29.4 | 36.7 | 31.6 | 28.0 | 32.5 | 33.8 | 27.0 | 26.1 | 32.1 | 57.9 | 53.3 |
| HB | 71.7 | 73.2 | 75.6 | 72.2 | 77.1 | 72.7 | 75.6 | 76.1 | 77.1 | 80.5 | 75.6 | 74.6 | 73.7 | 73.7 | 71.2 | 77.6 | 75.1 | 75.1 | 78.0 | 77.6 | 76.6 |
| JV | 94.9 | 86.5 | 96.2 | 79.7 | 98.1 | 98.4 | 98.9 | 98.7 | 97.8 | 98.9 | 98.4 | 96.2 | 99.2 | 98.4 | 95.9 | 98.9 | 96.2 | 96.2 | 98.4 | 99.2 | 97.0 |
| PS | 71.1 | 98.3 | 75.1 | 39.9 | 86.7 | 86.1 | 68.8 | 82.1 | 82.7 | 81.5 | 83.2 | 82.7 | 87.3 | 80.9 | 86.0 | 83.8 | 75.1 | 88.4 | 89.6 | 83.8 | 90.2 |
| SCP1 | 77.7 | 84.6 | 90.8 | 68.9 | 84.0 | 90.8 | 84.6 | 92.2 | 90.4 | 90.1 | 88.1 | 84.0 | 89.4 | 88.7 | 89.6 | 92.5 | 87.3 | 89.8 | 91.8 | 91.8 | 92.5 |
| SCP2 | 53.9 | 48.9 | 53.3 | 46.6 | 52.8 | 52.2 | 55.6 | 53.9 | 56.7 | 53.3 | 53.3 | 50.6 | 57.2 | 54.4 | 55.0 | 56.1 | 50.5 | 51.1 | 57.2 | 61.7 | 66.7 |
| SA | 96.3 | 69.6 | 71.2 | 31.9 | 100.0 | 100.0 | 95.6 | 98.4 | 97.0 | 100.0 | 100.0 | 100.0 | 100.0 | 100.0 | 100.0 | 98.8 | 81.4 | 100.0 | 99.0 | 99.9 | 99.0 |
| UG | 90.3 | 75.9 | 94.4 | 41.2 | 87.8 | 85.9 | 88.4 | 85.6 | 85.6 | 85.6 | 83.4 | 85.9 | 87.5 | 85.3 | 85.0 | 86.6 | 82.1 | 80.3 | 85.3 | 91.3 | 93.8 |
| Avg. | 67.0 | 66.0 | 72.5 | 48.6 | 71.8 | 70.9 | 70.3 | 71.9 | 71.5 | 72.1 | 70.8 | 71.1 | 72.7 | 70.7 | 71.0 | 73.0 | 67.5 | 70.4 | 73.6 | 76.04 | **80.05** |
| Rank | 15.20 | 15.55 | 10.25 | 19.55 | 10.40 | 11.70 | 12.40 | 9.40 | 9.95 | 8.90 | 12.80 | 11.50 | 7.30 | 12.40 | 12.85 | 6.45 | 14.60 | 12.65 | 6.40 | 6.40 | **4.35** |

size, epochs, and initial learning rate varied on the datasets for optimal performance. The hyperparameters were selected based on the Train and Test set provided by the dataset archive following TimesNet Wu et al. (2023). Moreover, the optimization was performed utilizing a cosine-annealing learning rate scheduler. We measure the prediction performance of our method using accuracy metric where larger values indicate better prediction accuracy.

**Baseline Models**   In this study, we evaluate the performance of our proposed method, TSCMamba, against ~~19~~20 state-of-the-art baselines in Table 1, encompassing Transformer-based (Eldele et al., 2024; Wu et al., 2023; Vaswani et al., 2017; Kitaev et al., 2020; Zhou et al., 2021; Liu et al., 2021; Wu et al., 2021; Liu et al., 2022; Zhou et al., 2022; Woo et al., 2022; Wu et al., 2022), CNN-based (Franceschi et al., 2019), RNN-based (Hochreiter & Schmidhuber, 1997; Lai et al., 2018; Gu et al., 2022), MLP-based (Zeng et al., 2023; Zhang et al., 2022a), and classical machine learning-based methods (Berndt & Clifford, 1994; Chen & Guestrin, 2016; Dempster et al., 2020). Therefore the comparison among these methods following Times-Net (Wu et al., 2023) provides strong recent baselines from various aspects of machine learning.

## 4.3   Predictive Performance Comparison

The comprehensive results are presented in Table 1. Notably, our approach achieves a substantial improvement of 4.01% over the existing best baseline, TSLANet (Eldele et al., 2024). Additionally, it improves upon the existing second-best baseline, TimesNet (Wu et al., 2023), by 6.45%, which is a significant margin compared to TimesNet's improvement of 0.6% over the previous best baseline, Flowformer (Wu et al., 2022). This notable performance gain establishes TSCMamba as a strong contender for the TSC task. ~~We plan to release our code and checkpoints for Table 1 publicly.~~ We have uploaded our code in supplementary materials and plan to release it publicly. While Table 1 presents our best-achieved results, we also demonstrate TSCMamba's reproducibility and stability across 5 runs with mean and error bars (standard deviation) in S-Figure 7. In addition to the main baselines, we also compare TSCMamba against regular Mamba (S-Figure 4). From Table 1, it is evident that some methods may perform well on certain datasets (e.g., TimesNet (Wu et al., 2023) on JV, SA), but may lack performance by a large margin on others (e.g., TimesNet (Wu et al., 2023) on EC, HW). In contrast, our method maintains a balance across the datasets while showing a significant improvement in average performance.

In addition to benchmark datasets mentioned by TimesNet (Wu et al., 2023), we also evaluate our model on 10 additional randomly selected datasets from TSLANet (Eldele et al., 2024) and present the results in S-Table 2. Following TimesNet (Wu et al., 2023) and other recent baselines like TSLANet (Eldele et al., 2024), we present the results as average performance across datasets. However, recognizing that averaging over various datasets might not be the optimal way to assess different models, we also present ranks of different models as suggested by Demšar (2006). These ranks are calculated using the Wilcoxon signed-rank test, following the methodology of Ismail Fawaz et al. (2019). For both sets of benchmark datasets, Table 1 and S-Table 2 demonstrate that TSCMamba achieves the best performance in terms of averaged accuracy and rank.

## 4.4   Computational Complexity

In this study, we compared the floating-point operations (FLOPs) of the top-performing methods presented in Table 1. To calculate FLOPs, we set a batch size of 16 across all baselines. For our method, we employed the best-performing hyperparameters, whereas for other baselines, we utilized the recommended parameters specified in the official TimesNet code (Wu et al., 2023) and Flowformer (Wu et al., 2022). We leveraged the source code from Ye (2023) to calculate FLOPs. The overall FLOPs, including both forward and backward passes, are presented in Table 2. Notably, our method achieves substantial improvements in terms of FLOPs across all datasets, with the exception of PEMS-SF (PS). This anomaly can be attributed to the projected space $(X)$ used to achieve the best result for this dataset, which was set to 1024, thereby impacting the total FLOPs for this dataset only.

Table 2: FLOPs comparison among the top performing methods. The values are represented in GigaFLOPS (G) or TeraFlops (T), where 1TFLOPs=1000GFLOPs and a lower value indicates better computational efficiency.

| Methods | EC | FD | HW | HB | JV | PS | SCP1 | SCP2 | SA | UG |
|---|---|---|---|---|---|---|---|---|---|---|
| Flow. Wu et al. (2022) | 1.06T | 37.97G | 92.21G | 246.37G | 15.76G | 94.02G | 542.64G | 697.74G | 50.33G | 190.82G |
| TimesNet. Wu et al. (2023) | 1.11T | 161.93G | 115.88G | 182.69G | 48.15G | **74.18G** | 503.62G | 2.33T | 26.00G | 247.73G |
| TSCMamba (Ours) | **1.69G** | **11.53G** | **27.24G** | **8.39G** | **12.33G** | 2.84T | **3.42G** | **11.11G** | **0.78G** | **13.86G** |

Table 3: Ablation experiments on particular components in our method.

| Mamba | Avg.Pool | ROCKET | AF | EC | FD | HW | HB | JV | PS | SCP1 | SCP2 | SA | UG | Avg. |
|---|---|---|---|---|---|---|---|---|---|---|---|---|---|---|
| ✓ | ✓ | ✓ | ✓ | 62.0 | 57.0 | 53.3 | 74.1 | 93.0 | 90.2 | 92.5 | 66.7 | 94.1 | 93.8 | 77.67 |
| ✗ | ✓ | ✓ | ✓ | 33.1 | 63.2 | 34.1 | 73.2 | 85.4 | 81.5 | 86.7 | 57.2 | 74.0 | 89.1 | 67.75 |
| ✓ | ✗ | ✓ | ✓ | 31.6 | 64.2 | 52.0 | 74.1 | 94.1 | 63.0 | 86.7 | 60.6 | 96.7 | 92.8 | 71.58 |
| ✓ | ✓ | ✗ | ✓ | 31.6 | 69.4 | 24.8 | 76.6 | 97.0 | 87.3 | 91.8 | 58.3 | 97.6 | 86.2 | 72.06 |
| ✓ | ✓ | ✓ | ✗ | 30.0 | 51.5 | 49.3 | 72.7 | 91.4 | 84.4 | 88.7 | 58.9 | 90.0 | 90.3 | 70.72 |

## 5 Ablation

### 5.1 Component-wise Ablation

In this section, we conduct an ablation study to investigate the contribution of individual components in our proposed method. The results are presented in Table 3. A notable observation is that the performance degrades in a large margin across all datasets when the Mamba modules are not utilized, highlighting the importance of incorporating Mamba in our approach. Specifically, when Mamba is not employed (2nd row), the intermediate values are bypassed by scanning operations and directly fed into the DP and MLPs for class logit prediction. In the 3rd row, we present depth-wise max-pooling instead of average-pooling, resulting in an input shape of B,3X. Furthermore, when ROCKET-extracted features are not utilized (4th row), we resort to MLP-extracted features, where the former are non-learnable and the latter are learnable. Additionally, in the 5th row, we explore the effect of replacing additive fusion (AF) with multiplicative fusion (MF) (of ROCKET features and spectral features), as detailed in Sec.3.3. Notably, while Table 3 largely mirrors the best performance reported in Table 1 across most datasets, the SpokenArabicDigits (SA) dataset exhibits optimal performance when employing depth-wise max-pooling and MLP-based features.

### 5.2 Hyperparameter Sensitivity

In this section, we discuss the key settings for the Mamba model, as detailed in Gu & Dao (2023). The model operates with four main settings: model dimension (d_model), SSM state expansion factor (d_state), local convolution width (d_conv), and block expansion factor (expand). In our experiments, we automatically set the model dimension based on the input data, while we adjust the other three settings. The importance of fine-tuning these parameters is evident from our tests, shown in Figure 2, which clearly demonstrate their impact on our model's performance. In addition to Mamba's hyperparameters, we also tuned the dimension of the projected space ($X$) mentioned in 1.

### 5.3 Effectiveness of Tango Scanning

Although, our way of scanning (Tango Scanning) at first glance may seem counterintuitive to only forward-based scanning and with another additional reverse flip based scanning (Schiff et al., 2024), it provides substantial improvements in accuracy. This approach, as demonstrated its effectiveness by Figure 3, outperforms traditional forward-scanning and other reverse-flip-based scanning for TSC tasks, making it a valuable strategy for complex scanning scenarios. Our tango scanning is explained in more detail in section 3.4.

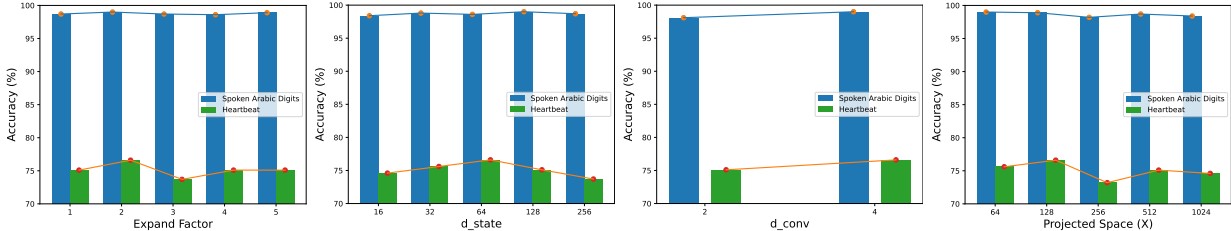

Figure 2: Sensitivity analysis of TSCMamba's hyper-parameters on Time Series Classification (TSC) performance. The plot shows the impact of varying (from left to right, top to bottom) block expansion factor, SSM state expansion factor, local convolutional width, and dimension of the projected space (X) on model performance, highlighting the relative importance of each component in achieving optimal TSC results.

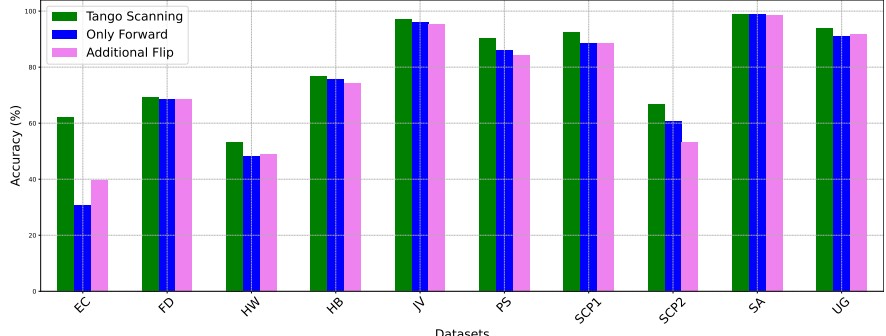

Figure 3: Effectiveness of our tango scanning compared against only forward-based scanning protocol and additional flip-based scanning protocol in reverse scanning.

## 6 Conclusion and Future Work

We present TSCMamba, an innovative approach for Time Series Classification (TSC) designed to enhance performance while maintaining lower Floating Point Operations (FLOPs). TSCMamba leverages multi-view learning to analyze different views of time-series data, including local and/or global features extracted from time and frequency domains, thereby capturing the essential, discriminative patterns of real-world time-series data. Moreover, the proposed tango scanning mechanism demonstrates TSCMamba's superiority over conventional scanning methods through extensive experimental validation. Our comprehensive experiments highlight TSCMamba's exceptional performance in terms of both accuracy and computational efficiency, consistently outperforming current state-of-the-art methods. These results suggest that TSCMamba can serve as a robust and efficient solution for a wide range of TSC applications. Looking ahead, our future work will focus on further enhancing TSCMamba by incorporating self-supervised learning techniques and extending its capabilities to multiple-task learning, in addition to the classification task. We also plan to explore the adaptability of TSCMamba across more diverse and complex time-series datasets, aiming to establish its versatility and robustness in various real-world scenarios. The promising results suggest TSCMamba's high potential for time series applications.

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

# 7 Appendix

In the appendix section, we present additional supplementary materials including algorithms, dataset statistics, etc. To distinguish from original materials, we add the prefix **S-** to the supplementary materials.

## 7.1 Algorithms

In this section, we present the two schematic algorithms for conversion of raw signals to CWT and ROCKET feature extraction in S-Algorithm 1 and S-Algorithm 2 respectively.

---
**S-Algorithm 1** Convert Raw Signals to CWT representation

---
**Input:** Raw signals of shape $(N, D, L)$
**Output:** Tensor of shape $(N, D, L_1, L_1)$ # We set $L_1 = 64$ in this paper.
 1: **for** each signal $i$ in $N$ **do**
 2:     **for** each dimension $d$ in $D$ **do**
 3:         $signal \leftarrow \text{Raw}[i, d, :]$
 4:         $coeff, freq \leftarrow \text{CWT}(signal)$
 5:         $cwt\_resized \leftarrow \text{resize}(coeff, (L_1, L_1), \text{mode="constant"})$
 6:         $\text{Tensor}[i, d, :, :] \leftarrow cwt\_resized$
 7:     **end for**
 8: **end for**

---

---
**S-Algorithm 2** Feature Extraction with ROCKET for Random Convolutional Kernel Transform

---
**Input:** Time series data of length $= L$, number of kernels, $n = \frac{X}{2}$
**Output:** Feature vector of shape $(2 \times n) = X$
 1: $kernels \leftarrow$ list of $n$ random kernels of random length $l$, weight $w$, bias $b$, dilation $d$, padding $p$.
 2: $feature\_maps \leftarrow$ empty list
 3: **for** each kernel $k$ in $kernels$ **do**
 4:     $h_{ppv}, h_{max} \leftarrow \text{convolve}(k, x)$
 5:     $feature\_maps.\text{append}(h_{ppv}, h_{max})$
 6: **end for**
 7: **return** feature_ maps

---

## 7.2 Dataset Statistics

We utilized ~~10~~ a total of 20 datasets following TimesNet (Wu et al., 2023) and TSLANet (Eldele et al., 2024). Datasets with their corresponding number of channels ($D$), sequence length, Train samples, Test samples, number of classes, and domain information are presented in S-Table 1.

S-Table 1: Publicly available datasets with their statistics utilized in this paper.

|  | Datasets | Channels | Length | Train | Test | Classes | Domain |
|---|---|---|---|---|---|---|---|
| Benchmark datasets | EthanolConcentration (EC) | 3 | 1751 | 261 | 263 | 4 | Alcohol Industry |
|  | FaceDetection (FD) | 144 | 62 | 5890 | 3524 | 2 | Face (250Hz) |
|  | Handwriting (HW) | 3 | 152 | 150 | 850 | 26 | Smart Watch |
|  | Heartbeat (HB) | 61 | 405 | 204 | 205 | 2 | Clinical |
|  | JapaneseVowels (JV) | 12 | 29 | 270 | 370 | 9 | Audio |
|  | PEMS-SF (PS) | 963 | 144 | 267 | 173 | 7 | Transportation |
|  | SelfRegulationSCP1 (SCP1) | 6 | 896 | 268 | 293 | 2 | Health (256Hz) |
|  | SelfRegulationSCP2 (SCP2) | 7 | 1152 | 200 | 180 | 2 | Health (256Hz) |
|  | SpokenArabicDigits (SA) | 13 | 93 | 6599 | 2199 | 10 | Voice (11025Hz) |
|  | UWaveGestureLibrary (UG) | 3 | 315 | 120 | 320 | 8 | Gesture |
| Additional datasets | AtrialFibrillation (AF) | 2 | 640 | 15 | 15 | 3 | ECG |
|  | BasicMotions (BM) | 6 | 100 | 40 | 40 | 4 | Human Activity Recognition |
|  | Cricket (CR) | 6 | 1197 | 108 | 72 | 12 | Human Activity Recognition |
|  | FingerMovements (FM) | 28 | 50 | 316 | 100 | 2 | EEG |
|  | HandMovementDirection (HMD) | 10 | 400 | 160 | 74 | 4 | EEG |
|  | MotorImagery (MI) | 64 | 3000 | 278 | 100 | 2 | EEG |
|  | PenDigits (PD) | 2 | 8 | 7494 | 3498 | 10 | Motion |
|  | PhonemeSpectra (PHS) | 11 | 217 | 3315 | 3353 | 39 | Audio |
|  | RacketSports (RS) | 6 | 30 | 151 | 152 | 4 | Human Activity Recognition |
|  | StandWalkJump (SWJ) | 4 | 2500 | 12 | 15 | 3 | ECG |

## 7.3 Comparison with Mamba

In addition to the robust baselines presented in this paper, we also evaluate the performance of Mamba (Gu & Dao, 2023). To conduct this experiment on the 10 benchmark datasets listed in Table 1, we process the MTS data directly using Mamba modules with a regular scanning protocol. The processed signals are then fed into two-stage MLPs, following the strategy outlined in Figure 1, to obtain class logits. The experimental results, shown in S-Figure 4, clearly demonstrate the effectiveness of our method (TSCMamba), as it outperforms all the benchmarks across the 10 datasets, underscoring the necessity of our approach. These experiments were conducted using the same hyperparameters as those used to achieve the results in Table 1.

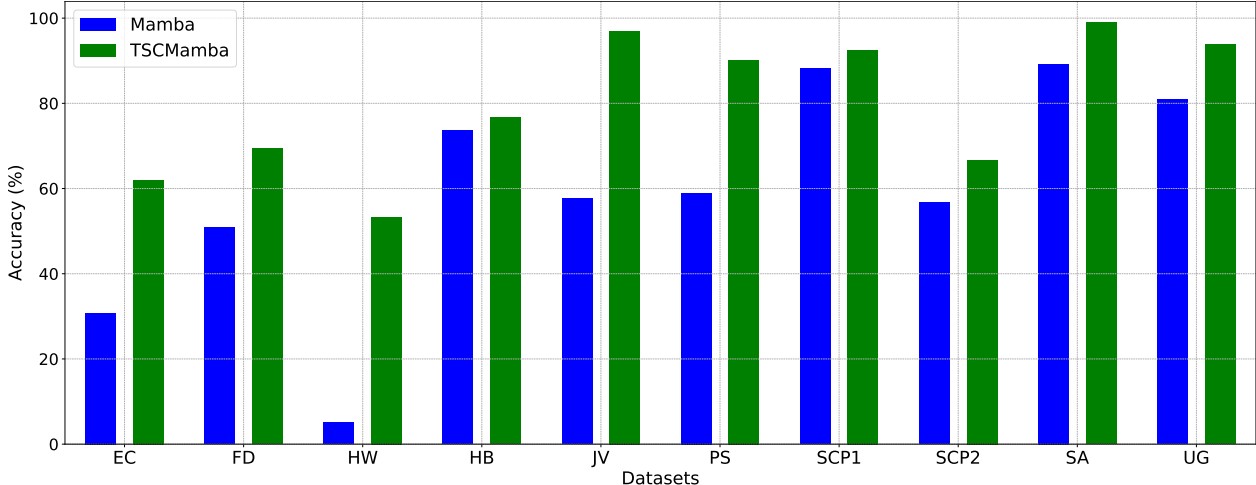

S-Figure 4: TSCMamba in comparison with directly applied regular Mamba module to time series data

## 7.4 Comparison with BiMamba

In this section, we present the comparative results of TSCMamba's channel and token (or time) tango scanning (Eqs. 12 and 11) against BiMamba (Schiff et al., 2024). To ensure a fair comparison, we utilized the same hyperparameters that were employed to achieve the best results shown in Table 1. For BiMamba, we used the official code provided by Schiff et al. (2024), with tied weights for the in and out projections. As illustrated in S-Figure 5, TSCMamba consistently outperforms BiMamba across all benchmark datasets, underscoring the effectiveness of tango scanning.

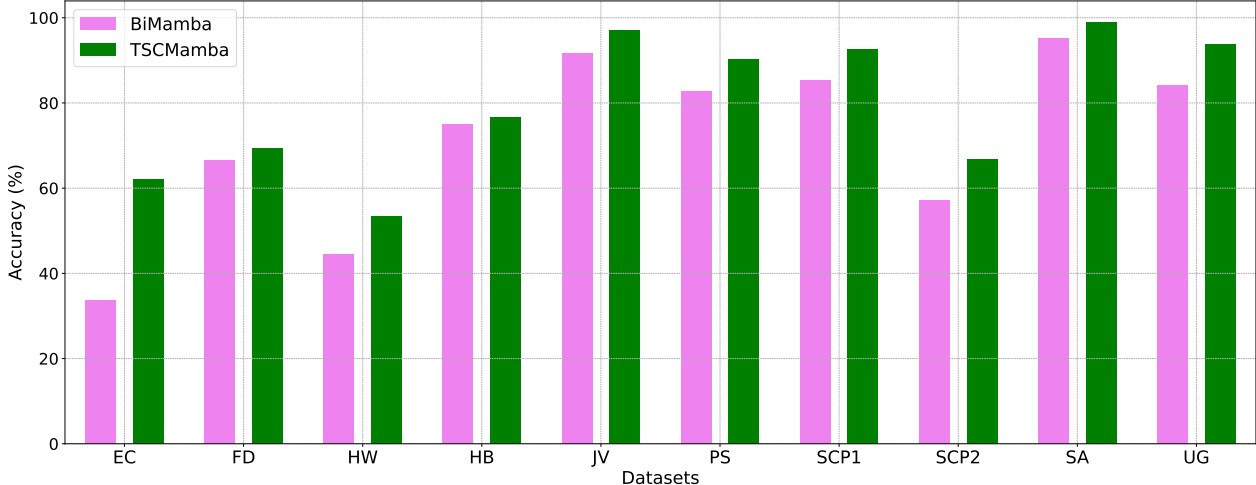

S-Figure 5: Performance comparison in accuracy of TSCMamba with tango scanning versus BiMamba (Schiff et al., 2024) across the benchmark datasets.

## 7.5 Ablation on Tango Scanning

This section presents the results of our experiments with both channel and token tango scanning. We display the outcomes of token tango scanning with the channel tango scanning module turned off. Conversely, we also provide results for channel tango scanning with the token tango scanning module disabled. As demonstrated in S-Figure 6, both modules are crucial for achieving optimal accuracy, highlighting their individual importance in the overall performance of our model.

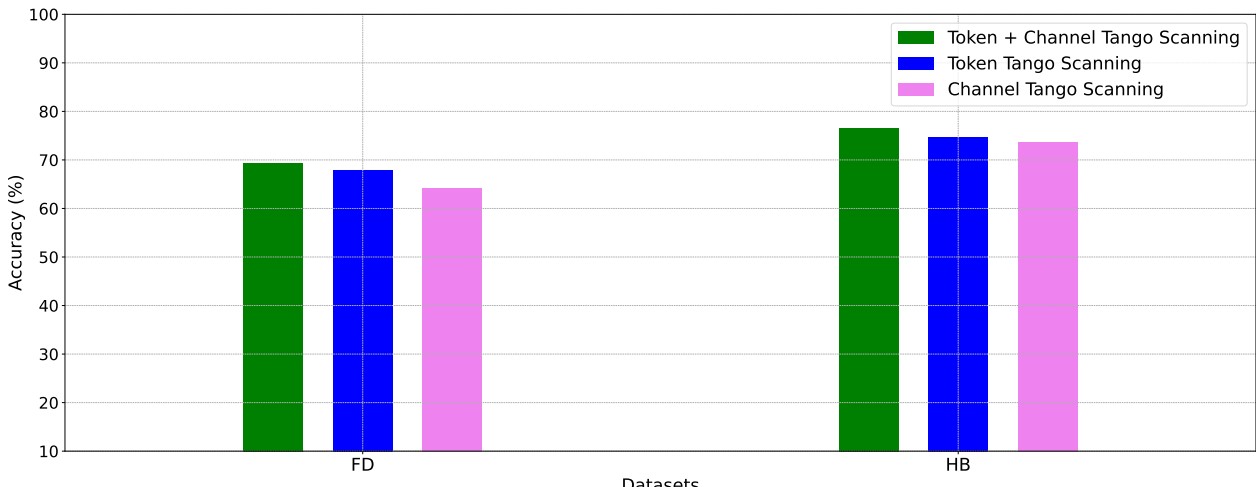

S-Figure 6: Ablation on tango scanning module compared against only token tango scanning and only channel tango scanning.

## 7.6 Accuracy on Multiple Runs

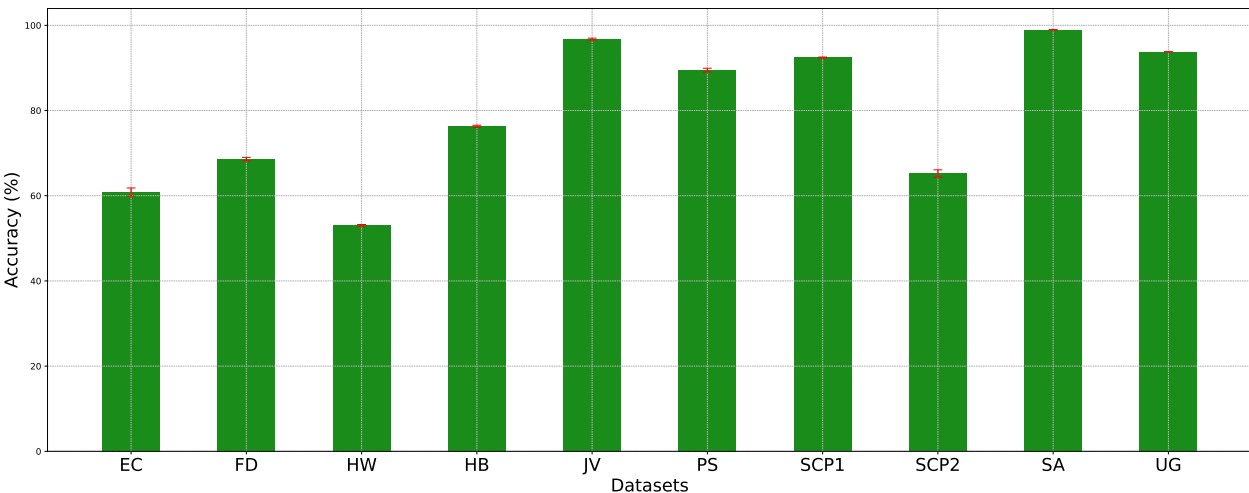

S-Figure 7: Performance of TSCMamba over 5 random runs. The mean performance is shown as green bars, with the standard deviation represented by red error bars that are very small. (Best viewed when zoomed in.)

## 7.7 Additional datasets results from UEA

S-Table 2: Additional classification results on the UEA datasets in terms of accuracy (as %). The ranks are calculated using the Wilcoxon signed-rank test (lower is better).

| Dataset | TSCMamba Ours | TSLANet (2024) | GPT4TS (2023) | TimesNet (2023) | ROCKET (2020) | CrossF. (2023) | PatchTST (2023) | MLP (2023) | TS-TCC (2021) | TS2VEC (2022) |
|---|---|---|---|---|---|---|---|---|---|---|
| AtrialFibrillation | **67.00** | 40.00 | 33.33 | 33.33 | 20.00 | 46.66 | 53.33 | 46.66 | 33.33 | 53.33 |
| BasicMotions | **100.00** | 100.00 | 92.50 | 100.00 | 100.00 | 90.00 | 92.50 | 85.00 | 100.00 | 92.50 |
| Cricket | **98.61** | 98.61 | 8.33 | 87.50 | 98.61 | 84.72 | 84.72 | 91.67 | 93.06 | 65.28 |
| FingerMovements | **69.00** | 61.00 | 57.00 | 59.38 | 61.00 | 64.00 | 62.00 | 64.00 | 44.00 | 51.00 |
| HandMovementDirection | **71.62** | 52.70 | 18.92 | 50.00 | 50.00 | 58.11 | 58.11 | 58.11 | 64.86 | 32.43 |
| MotorImagery | **62.00** | 62.00 | 50.00 | 51.04 | 53.00 | 61.00 | 61.00 | 47.00 | 47.00 | 47.00 |
| PenDigits | 98.54 | 98.94 | 97.74 | 98.19 | 97.34 | 93.65 | **99.23** | 92.94 | 98.51 | 97.40 |
| PhonemeSpectra | 24.66 | 17.75 | 3.01 | 18.24 | 17.60 | 7.55 | 11.69 | 7.10 | **25.92** | 8.23 |
| RacketSports | **91.45** | 90.79 | 76.97 | 82.64 | 86.18 | 81.58 | 84.21 | 78.95 | 84.87 | 74.34 |
| StandWalkJump | **73.33** | 46.67 | 33.33 | 53.33 | 46.67 | 53.33 | 60.00 | 60.00 | 40.00 | 46.67 |
| Average | **75.62** | 66.85 | 47.11 | 63.36 | 63.04 | 64.06 | 66.68 | 64.54 | 63.15 | 56.82 |
| Rank | **1.65** | 3.90 | 8.60 | 5.70 | 5.70 | 6.00 | 4.35 | 5.95 | 5.45 | 7.70 |

