# OpenReview forum: "TSCMamba: Mamba Meets Multi-View Learning for Time Series Classification"
_TMLR — Rejected by TMLR_

### Review · Reviewer_TYFS · 2024-07-08

**Summary Of Contributions:**

The paper aggregates techniques for feature extraction from time-series data and combines them with the Mamba SSM to extract quantities of interest. The authors describe a bidirectional SSM that leads to a reversal invariance. The authors validate their findings on common time-series classification datasets.

**Audience:**

Yes

**Broader Impact Concerns:**

I have no concerns.

**Claims And Evidence:**

No

**Requested Changes:**

### Contribution Claims are too Strong
- "However, effective multi-view strategies have not been well investigated for enhancing deep learning-based TSC in the literature." There are models like [3] and the references therein that split the time series into "global" and "local" features.
- "Moreover, we propose an innovative Mamba-based scanning scheme, called tango scanning", c.f. bidirectional SSM [1], [2], [4]
- "This scanning is demonstrated to be more effective in modeling the relationships in the sequence than that of vanilla Mamba block for TSC" - I did not find a result in the paper that backs this claim.
- "We first use the CWT for multi-scale representation of MTS in the general task of TSC.", c.f. [5] which uses the wavelet transform for TSC.
### Back arguments by reference or experiment.
- Wavelets are local in frequency and time domain.
- The paper presents engineering choices and challenges without proper reasoning. In example:
  - p.7: "If the history tokens do not contain informational contexts, Mamba may provide less effective
predicted output. To alleviate this potential limitation of causal scanning, we construct a dedicated module
to extend a vanilla Mamba block." - It is unclear how reverting the sequence can add information.
  - p.8: "Performing Tango Scanning in Time and Channel Dimension" - How you arrived at this design choice is unclear. There are no references nor experimental ablations.
### Release Code.
While the authors write in section 4.3 that they plan to release code, the results are not reproducible without implementation details. The paper's main contribution is in aggregating time-series tools into a feature extraction framework for classification that performs well on datasets. Without the actual implementation, I find it hard to justify that this is interesting to the TMLR community.
### Bidirectional SSM
- The proposed "Tango Scanning" is not novel in the sense the authors describe it.
  - "Bidirectional" SSM are well known, c.f. [1], [2], [4]. The difference to, e.g. [Schiff et al. (2024)][1] is that the output is not reversed, but, e.g. [4] uses non-reversed outputs but without weight tying.
  - The proposed method is enforcing a reversibility invariance, i.e. $f(x, t)=f(x, \text{reverse}[t])$, whereas typically a reversability equivariance is enforced $ \text{reverse}[f(x, t)]=f(x,  \text{reverse}[t])$, c.f. [1]. I am unaware of prior work enforcing this constraint, possibly because it might be restrictive in a forecasting setting. [4] claims that an advantage of bidirectional SSMs is their resistance to noise.
- Bidirectional scanning is not ablated. It is not immediate to me why bidirectional scanning would have an edge over forward scanning only.

### Further Comments
- Table 3: no std. deviations
- p.8: The SSM in [Schiff et al. (2024)][1] are weight-tied, meaning that they are the same SSM.

[1]: https://arxiv.org/abs/2403.03234
[2]: https://arxiv.org/abs/2403.11144v3
[3]: https://proceedings.neurips.cc/paper_files/paper/2023/hash/28b3dc0970fa4624a63278a4268de997-Abstract-Conference.html
[4]: https://arxiv.org/abs/2402.08678
[5]: https://ieeexplore.ieee.org/abstract/document/1614807

**Strengths And Weaknesses:**

Strengths
- The proposed method combines time-series techniques into a single model.
- The reported experimental performance is good.

Weaknesses
- The paper is not reproducible.
  - No code is published.
  - Arguments are frequently presented without backing (experiment/ proof/ reference).
- It is unclear why the proposed "Tango Scanning" should perform well.
- Some related work that applies similar design principles is not referenced.

---

> ### Author Response · Authors · 2024-08-17
> **Response to Reviewer TYFS (1/2)**
>
> Thank you for your valuable suggestions and comments. We greatly appreciate the suggestions. We address the concerns below:
>
> - **Code is not published:**
> Thanks for your interest. We have uploaded the code with the logs for the main results in supplementary files.
> - **Arguments without backing:**
> Thanks for your comment. We have added the references in our revised paper and more experimental verifications (e.g.: additional datasets evaluation, direct comparison with Mamba, etc.). For details please see below at "Back arguments by reference or experiment''.
>
> - **Why the proposed "Tango Scanning" should perform well**:
> Thanks for your question. We have experimentally verified our method compared against regular (only forward) and bi-directional scanning (with an additional flip) in Figure 3. Our Tango Scanning is inversion invariant capturing more intrinsic patterns for time series classification. By identifying patterns meaningful in both directions, the TSC model may focus more on intrinsic patterns while being less affected by (potentially direction-specific) noise. Additionally, in S-Figure 6 we have demonstrated the effectiveness of Tango Scanning via ablation study.
>
> - **Some related work that applies similar design principles is not referenced**:
> Thanks for your suggestion. We have updated our related work sections with additional proper references, including all the suggested references.
>
>
> ## Requested Changes
> ### Contribution Claims are too Strong
> - **"However, effective multi-view strategies have not been well investigated for enhancing deep learning-based TSC in the literature." There are models like [3] and the references therein that split the time series into "global" and "local" features:** Thank you for your concern. We revised the corresponding section in our revised paper and removed this line. However, Koopa [3] is dedicated to forecasting tasks, while ours is dedicated to time-series classification (TSC) tasks.
>
> - **...innovative Mamba-based scanning scheme, called tango scanning...**: In our revised paper, we have added a section compared against BiMamba [1] in Supplementary Section 7.4. In [1], only the in and out projections are weight-tied between two Mamba modules, leaving the SSM modules to be different. Our method involves only one Mamba enforcing inversion invariance, requiring smaller memory. [2] have used separate Mamba modules for forward and reverse scanning. [4] also used separate weights for two-directional scanning (Please see their paper's paragraph between equations 8 and 9). Please see our discussion in the last paragraph on page 3 of our revised paper.
>
> - **This scanning is demonstrated to be more effective in modeling the relationships in the sequence than that of vanilla Mamba block for TSC**: Thank you for your comment. We have added experimental verification in our Supplementary Section 7.3 (Comparison with Mamba).
> - **We first use the CWT for multi-scale representation of MTS in the general task of TSC**: We have removed this claim and also added the reference [5]. Thanks for your suggestion.
>
> ### Back arguments by reference or experiment.
> - **Wavelets are local in frequency and time domain.**: We have added references. Please see page 2 of our paper for more details.
>
> - **p.7: "If the history tokens do not contain ... It is unclear how reverting the sequence can add information**: Thanks for your comment. We have discussed the inversion invariance property and discussed why it may help with time series classification. Please see around the middle part of page 3.
>
> - **p.8: "Performing Tango Scanning in Time and Channel Dimension" - How you arrived at this design choice is unclear:** Please see the discussion on tango scanning in Subsection 3.4. Moreover, to further verify via experimentally, we have added results in our Supplementary. Please see Subsection 7.5 Ablation on Tango Scanning.
>
> ### Release Code
> We have uploaded the code to supplementary materials.
>
> ### Bidirectional SSM
> - **The proposed "Tango Scanning" is not novel in the sense the authors describe it.**: Thank you for your comment. We have improved our presentation. In particular, we have clarified the differences from [1], [2], and [4]. Please see our discussion on page 3, last paragraph, and also Supplementary Subsection 7.4 (Comparison with BiMamba).
>
> - **"Bidirectional" SSM are well known...**: Please see our response to the comment above.
>
> - **... enforcing a reversibility invariance...reversability equivariance...**: Thank you for your insightful comment. We agree with you on this. We have discussed inversion invariance in more detail on page 3, around the middle part.
>
> - **Bidirectional scanning is not ablated**: Thanks for your suggestion. We have added more ablation studies. Please see Figure 3, Supplementary Subsections 7.4 and 7.5

---

> > ### Author Response · Authors · 2024-08-17
> > **Response to Reviewer TYFS (2/2)**
> >
> > ### Further Comments
> > - **Table 3: no std. deviations**: Thank you for your comment. We agree that in that Table we did not provide the std. because those were meant to be component-wise ablation studies for our method. However, we performed study on the variance of our method. The standard deviation for different datasets is typically very small. Please see Supplementary S-Figure 7.
> >
> > - **p.8: The SSM in Schiff et al. (2024) are weight-tied, meaning that they are the same SSM.**: Please see our response to Bidirectional SSM (1st comment).
> >
> > [1] Yair Schiff, Chia-Hsiang Kao, Aaron Gokaslan, Tri Dao, Albert Gu, and Volodymyr Kuleshov. Caduceus:
> > Bi-directional equivariant long-range dna sequence modeling. arXiv preprint arXiv:2403.03234, 2024
> >
> > [2] Wang, Zihan, et al. "Is Mamba Effective for Time Series Forecasting?." arXiv preprint arXiv:2403.11144 (2024).
> >
> > [3] Liu, Yong, Chenyu Li, Jianmin Wang, and Mingsheng Long. "Koopa: Learning non-stationary time series dynamics with koopman predictors." Advances in Neural Information Processing Systems 36 (2024).
> >
> > [4] Ali Behrouz and Farnoosh Hashemi. Graph Mamba: Towards learning on graphs with state space models.
> > arXiv preprint arXiv:2402.08678, 2024.
> >
> > [5] Qibin Zhao and Liqing Zhang. Ecg feature extraction and classification using wavelet transform and support
> > vector machines. In 2005 International Conference on Neural Networks and Brain, volume 2, pp. 1089–
> > 1092, 2005. doi: 10.1109/ICNNB.2005.1614807.

---

> > > ### Comment · Reviewer_TYFS · 2024-09-04
> > > **Additional Comments**
> > >
> > > The paper has taken a positive trajectory in terms of backing the arguments with evidence and specifying contributions. Yet, introducing the concept of equivariance and making it central to the story introduced new unjustified statements.
> > >
> > > I summarized my remaining comments below and hope they help you in improving your manuscript.
> > >
> > > ## Major Comments
> > > - p.1 l.3: "shift equivariance and inversion equivariance are largely unexplored": there is literature addressing both, e.g. [1](https://openreview.net/forum?id=ga5SNulYet)
> > > - p.3 "Using inversion invariance..." Equivariance (and invariance as a special case) has been interpreted as a data augmentation technique in the geometric deep learning community, e.g. [2](https://openreview.net/forum?id=phnN1eu5AX) and its generalization analyzed e.g. in [3](https://proceedings.neurips.cc/paper_files/paper/2023/hash/adf5a38a2e2e7606fbfc3eff72998afa-Abstract-Conference.html). In this framework "Tango scanning" would be a reverse-invariant layer with mamba backbone since $[\operatorname{reverse}, \operatorname{id}]$ form a two-element group.
> > > - It is *still* unclear why the proposed "Tango Scanning" should perform well: from my understanding, the fused feature tensor $\mathcal{V}_W$ does not admit temporal structure anymore as B=batch, D=channel-dimension, X=feature-dimension. If this is the case, the statement p.1 "such as shift equivariance and inversion invariance," seems off as the inversion invariance is not time-series specific
> > > - p.11 "The hyperparameters were selected based on the Train and Test set" [the paper you are referencing](https://openreview.net/forum?id=ju_Uqw384Oq) has a train, validation, and test set. I guess that this is a typo; if not, the experimental results cannot be compared, and the improvements are probably void.
> > >
> > > ## Minor Comments
> > > - p.1 "Shift equivariance property is" grammar
> > > - p.3 "we propose a switch mechanism" It might be good to briefly sketch how it works here
> > > - p.4 "recursive neural networks (RNN)" recurrent? LSTM and GRU are recurrent
> > > - p.7 "In this paper, this is either" to focus on your final architecture can move multiplicative fusion to the supplementary and directly state you are using the additive one; from my understanding, all experiments except for the ablation are carried out with it
> > > - p.9 " hence call it tango scanning" I dislike the catchy name since it is not descriptive of what is going - it could be bi-directional scanning or anything else. As you built in inversion invariance into the manuscript, you could also term it like that.
> > > - p.11 "using accuracy method" grammar
> > > - p.13 Based on Figure 3, the proposed scanning seems to have a big impact on some tasks and seems to be unimportant in others. Are there structural differences/ similarities in those tasks? Does it depend on whether ROCKET is active or not?
> > > - p.20 The new empirical studies seem weak: Why are there only two datasets instead of 10?
> > > - tables 1, 2, 3, and S-2 are not consistent:  sometimes there are no highlights, and sometimes there are identical ACC scores, but one method (yours) and none of the tied methods are highlighted. I reckon this is due to truncation, but it isn't apparent, and the difference is hardly statistically significant. I prefer the style of table s-2 (with more ties) for all tables.

---

### Review · Reviewer_ouYB · 2024-07-09

**Summary Of Contributions:**

The authors introduce a multi-view time-frequency neural network architecture designed for multivariate time series classification. This framework leverages wavelet transform to break down the signal into its spectral and temporal components. It then uses a combination of CNN (inspired by the ROCKET method) and MLP to extract local and global features, respectively. To identify discriminative patterns in the temporal domain, the authors propose a switch mechanism that selects between local and global temporal patterns, integrating them with CWT features. For capturing long-term dependencies efficiently, the state-of-the-art state-space method, Mamba, introduced in 2023, is utilized due to its linear time complexity.

**Audience:**

Yes

**Claims And Evidence:**

Yes

**Requested Changes:**

Based on the weaknesses above the following changes can significantly improve the papers main content and contribution.
- **Supporting Motivation and Highlighting Contribution:** The introduction should be significantly improved to better convey the main message of the study and position the contribution of the proposed architecture, including specific references that confirm mentioned comments. (Refer to **W1,W2**)
- **Experimental Evaluation:** The experimental evaluation is poor and could be enhanced by more experiments as described above. Details on the reproducibility of the results and model variances are significant and should be provided at least in the supplementary material. (Refer to **W3,W4,W5,W6**)
- **Comparative Analysis:** The experimental evaluation could benefit from comparisons with methods that have similar components, in addition to the extensive comparisons with transformer-based architectures that are not specifically tailored to TSC but rather to forecasting. (Refer to **W3,W4,W7**)

**Strengths And Weaknesses:**

**Strengths:**
- **[S1]:** The authors integrate multiple advanced modules, such as time-frequency transforms, local and global neural network feature extraction, and the recent state-of-the-art Mamba architecture, to achieve high performance in multivariate time series classification. They also introduce a scanning scheme tailored to the Mamba architecture to identify salient features within complex contexts.
- **[S2]:** There is a notable performance improvement of approximately 6% in classification accuracy across 10 subsets of the UEA repository.
- **[S3]:** The proposed method demonstrates greater computational efficiency compared to baseline models.
- **[S4]:** The paper includes a thorough ablation study of the model components and various scanning techniques.

**Weaknesses:**
- **[W1]:** Traditional time series classification (TSC) methods that disentangle time and frequency components, including both older [1] and more recent works[2], have dominated the field. The authors should better clarify their choice of wavelet transform over other spectral transformation methods and include relevant works in the related work section. Additionally, several unsupported statements, particularly in paragraphs 2 and 4 on page 1, should be either referenced or removed.
- **[W2]:** The introduction section is very extensive and the paper's motivation is not clearly highlighted. There is a deviation from the current trend of general-purpose backbones and unsupervised pretraining tasks, as well as from testing on multiple time series tasks and datasets.
- **[W3]:** The experiments section is somewhat limited, as only 10 subsets from the UEA datasets (out of 26 in total) are selected. Other papers, including cited works, use additional datasets for classification and clustering. The authors should consider whether their architecture can be extended to other types of tasks and datasets (as in [2]).
- **[W4]:** The authors should compare their method directly to Mamba as a baseline since it is the component showing the most significant effect on the classification performance. It has also been recently incorporated in the Time Series Library [3,4].
- **[W5]:** In the experiments section, results appear identical to those in the original TimesNet paper. Random seeds and multiple runs to capture model variance are not taken into account.
- **[W6]:** The hyperparameter selection process for the proposed method and the baselines should be extensively detailed. The UEA datasets do not provide intermediate validation sets for parameter tuning, and some researchers validate and test on the test set, which may highlight models that overfit. If a validation set is used, this should be clearly stated in the experimental setup.
- **[W7]:** In the ablation experiments (Table 3), the average performance per combination should be mentioned to facilitate comparison with the baselines in Table 1.

[1] Zhang, X., Zhao, Z., Tsiligkaridis, T., & Zitnik, M. (2022). Self-supervised contrastive pre-training for time series via time-frequency consistency. Advances in Neural Information Processing Systems, 35, 3988-4003.

[2] Eldele, Emadeldeen, et al. "Tslanet: Rethinking transformers for time series representation learning." arXiv preprint arXiv:2404.08472 (2024).

[3] Wu, H., Hu, T., Liu, Y., Zhou, H., Wang, J., & Long, M. (2022). Timesnet: Temporal 2d-variation modeling for general time series analysis. arXiv preprint arXiv:2210.02186.

[4] https://github.com/thuml/Time-Series-Library

---

> ### Author Response · Authors · 2024-08-17
> **Response to Reviewer ouYB**
>
> Thank you for your valuable suggestion to improve our paper. We greatly appreciate the comments. We address the concerns below:
>
>
> - **[W1]** Thanks for your suggestion. We have discussed the suggested paper and also clarified further regarding our choice of wavelet transform.
> - **[W2]** Thanks for your comment. We have revised the introduction section to better convey the message.
> - **[W3]** Thanks for your suggestion. We followed the benchmark 10 datasets followed by TimesNet [1]. Our method is robust enough to incorporate additional datasets. Moreover, in our revised version we added an additional 10 datasets followed by TSLANet [2]. We presented the results in S-Table-2 of our paper.
> - **[W4]** Thanks for your suggestion. Mamba is included in the Time Series Library [3] for forecasting tasks. However, we followed your suggestion and added the comparative results on benchmark datasets while utilizing Mamba directly. The results are demonstrated in Supplementary Subsection 7.3.
>
> - **[W5]** Since TimesNet [1] was fine-tuned to achieve the best results, we did not re-tune the baselines to ensure a fair comparison and avoid any potential downgrade in their performance. Furthermore, the train and test sets were fixed across all baselines and our method, as provided by the dataset archive. This consistency ensures that our comparisons are fair and unbiased, reflecting the true performance differences between the methods.
>
> - **[W6]** We followed the strategy used in the TimesNet paper [1]. Following TimesNet [1], we did not use a separate validation set. The fixed train and test sets provided by the UEA dataset archive were used consistently across all experiments. In the revised paper, we have added this statement.
> - **[W7]** Thanks for your suggestion. We have added avg. performance per combination. However, we would like to further clarify that this table was meant to present the component-wise ablation of our method.
> ### Requested Changes:
> - **Supporting Motivation and Highlighting Contribution:** Thanks for your suggestion. We have updated our introduction section following the suggestions and tried our best to better convey the message. Since our updates include a major change, we did not separately use green/red colors for better visibility.
> - **Experimental Evaluation:** We have added more experimental results in our revised version following the suggestions. In our paper's S-Figure 7, we have demonstrated the model variances via 5 runs. For the reproducibility of the results, we have uploaded our code in supplementary material along with the logs file generated the main results regarding Table 1 and S-Table 2.
> - **Comparative Analysis:** We have added more results compared against TSLANet's [2] baselines and also we have added the performances of TSLANet [2] in our paper's Table 1. Thanks again for your valuable suggestion.
>
> [1] Haixu Wu, Tengge Hu, Yong Liu, Hang Zhou, Jianmin Wang, and Mingsheng Long. Timesnet: Temporal 2d-
> variation modeling for general time series analysis. In The Eleventh International Conference on Learning
> Representations, 2023. URL https://openreview.net/forum?id=ju_Uqw384Oq
>
>
> [2] Emadeldeen Eldele, Mohamed Ragab, Zhenghua Chen, Min Wu, and Xiaoli Li. Tslanet: Rethinking trans-
> formers for time series representation learning. In International Conference on Machine Learning, 2024
>
>
> [3] https://github.com/thuml/Time-Series-Library

---

### Review · Reviewer_YdVT · 2024-08-08

**Summary Of Contributions:**

The paper considers the time series classification problem. The standard approach using deep learning is to apply CNNs for local pattern-based features. On the other hand, the approach fails to capture global patterns. The paper proposes to use features based on wavelets of the time-series that can capture global patterns such as cyclic behaviours. More precisely, the proposed method also employs a state-space model to switching local feature-based CNNs and wavelets feature-based MLPs. The experimental results using benchmark datasets shows that the proposed method outperforms other baselines.

**Audience:**

Yes

**Claims And Evidence:**

Yes

**Requested Changes:**

As raised above, it is not clear how much the newly proposed components, the state-space model and the global features affects the performance. For example, to clarify the effect of the state-space model, the paper could compare the proposed methods with some simple baseline combining local and global features. If such additional experiments are performed, the technical contribution would be clearer.

**Strengths And Weaknesses:**

Strengths
- The paper shows a well-motivated approach to trying to capture the global features of time series.
-The experimental results show that the proposed method performs better than other alternatives.

Weaknesses:
-The effects or advantages of newly proposed components are not fully clear. In particular, how the state-space model and/or the global features based on wavelets should be clarified.

---

> ### Author Response · Authors · 2024-08-17
> **Response to Reviewer YdVT**
>
> Thanks so much for your valuable time. We appreciate your input and suggestions. Below we address the major concerns
> ## Requested Changes
> - **...it is not clear how much the newly proposed components, the state-space model and the global features affects the performance....:** In Table 3, we have performed several ablation studies regarding particular components. For example, please consider 2nd row, where we have demonstrated performance without considering the state-space-model (Mamba). Moreover, we have also compared against very recent baseline TSLANet [1] (Table 1). We have also added 10 other datasets in addition to the original benchmark 10 datasets for comparison in supplementary materials (S-Table 2). Also, to compare directly with state-space-machine (Mamba) we have added a separate Subsection in our supplementary material (7.3 Comparison with Mamba). Therefore, the state-space machine may not perform well, when directly applied to the Time Series Classification (TSC) task, which verifies the effectiveness of our proposed method TSCMamba.
>
> [1] Emadeldeen Eldele, Mohamed Ragab, Zhenghua Chen, Min Wu, and Xiaoli Li. Tslanet: Rethinking trans-
> formers for time series representation learning. In International Conference on Machine Learning, 2024

---

### Decision · Action_Editor_foPu · 2024-10-14

**Recommendation:** Reject

**Comment:**

This paper aims to address the task of multivariate time series classification (TSC) by focusing on underexplored properties such as shift equivariance and inversion invariance. The main drawback of this work is the lack of methodological innovation. The paper borrows heavily from existing methods, including spectral, temporal, local, and global features, as well as the Mamba-type model. However, the authors primarily use these methods without clearly explaining how they specifically address issues like shift equivariance and inversion invariance. The reviewers acknowledged the engineering contributions of the paper but also pointed out its research-related shortcomings. One of the reviews lacked sufficient substance and could not support its viewpoint in the final decision. In summary, this paper is not suitable for publication in an academic journal and would be more appropriate for dissemination as an engineering project or in promotional materials.

**Audience:**

This article holds certain value for researchers working in Multivariate time series classification (TSC) and multi-modal learning.

**Claims And Evidence:**

This paper primarily makes two claims. The first is related to its performance: "Experiments on two sets of benchmark datasets (10 datasets each) demonstrate our approach's effectiveness, achieving average accuracy improvements of 4.01-6.45% and 8.77%, respectively, over leading TSC models such as TimesNet and TSLANet." The experimental data are generally logical. The second claim involves some theoretical analysis of multivariate time series classification (TSC), including shift equivariance and inversion invariance. Unfortunately, the authors do not provide strong analysis or new insights. Additionally, they employ a variety of multimodal structures in an attempt to address the above analysis, but again, there is no supporting evidence.

**Resubmission Of Major Revision:**

The authors may consider submitting a major revision at a later time.